# Molecular Characterization and Haplotype Analysis of *Low Phytic Acid-1* (*lpa1*) Gene Governing Accumulation of Kernel Phytic Acid in Subtropically-Adapted Maize

Vinay Bhatt [1,2], Vignesh Muthusamy [1,*], Rashmi Chhabra [1], Ashvinkumar Katral [1], Shridhar Ragi [1], Vinay Rojaria [1], Gulab Chand [1], Govinda Rai Sarma [1], Rajkumar Uttamrao Zunjare [1], Kusuma Kumari Panda [2], Ashok Kumar Singh [1] and Firoz Hossain [1]

[1] Division of Genetics, ICAR-Indian Agricultural Research Institute, New Delhi 110012, India; vinaybhatt024@gmail.com (V.B.); reshu0428@rediffmail.com (R.C.); ashokgkatral@gmail.com (A.K.); shridharragi1996@gmail.com (S.R.); vvinay3878@yahoo.com (V.R.); gulab.biotech@yahoo.com (G.C.); raisarmag@gmail.com (G.R.S.); raj_gpb@yahoo.com (R.U.Z.); aks_gene@yahoo.com (A.K.S.); fh_gpb@yahoo.com (F.H.)

[2] AMITY Institute of Biotechnology, AMITY University, Noida 201313, India; pkkumari@amity.edu

\* Correspondence: pmvignesh@yahoo.co.in

**Abstract:** Maize is an important food, feed, fodder and industrial crop in addition to being a valuable source of micronutrients. Phytic acid (PA), an anti-nutritional factor in maize, makes crucial minerals inaccessible to monogastric animals. The *low phytic acid-1* (*lpa1*) gene located on chromosome-1S is 7292 bp long with 11 exons, and the recessive *lpa1-1* allele reduces the accumulation of PA thereby enhances the bioavailability of essential minerals in maize kernels. Here, we characterized the full-length *Lpa1* gene sequence in three mutants (*lpa1-1*) and seven wild-type (*Lpa1*) maize inbreds. Sequence analysis revealed 607 polymorphic sites across *Lpa1* sequences, indicating wide variability for *Lpa1* among the inbreds. Further, SNP from "C" to "T" differentiated wild-type and mutant-type alleles at 1432 amino acid position. Gene-based diversity among 48 diverse maize inbreds using 15 *InDel* markers revealed the formation of 42 distinct haplotypes; six of which (Hap6, Hap16, Hap17, Hap19 Hap27 and Hap31) were shared by more than one genotype. The number of exons in *Lpa1* ranged from 11–19 among maize genotypes and 6–14 among 26 orthologues. Major functional motifs of *Lpa1* detected were ATP-binding Cassette (ABC) transporter trans-membrane region and ABC transporter. Phylogenetic tree using nucleotide and protein sequences revealed a closer relationship of maize *Lpa1* sequences with *Sorghum bicolor*, *Panicum hallii*, *Setaria italica* and *S. viridis*. This study offered newer insights into the understanding of the genetic diversity of the *Lpa1* gene in maize and related crop-species, and information generated here would further help in exploiting the *lpa1* mutant for the enhancement of nutritional value in maize kernels.

**Keywords:** maize; low phytic acid; characterization; haplotype; gene-based diversity; *ZmMRP4*





## 1. Introduction

More than two billion people worldwide suffer from micronutrient deficiencies especially iron and zinc [1]. Iron is necessary for humans to maintain healthy red blood cells, muscles, and fundamental biological processes [2,3]. While, zinc plays a crucial part in the transcriptional control of the cellular metabolic network and is necessary for homeostasis, connective tissue development, growth, and maintenance, RNA and DNA synthesis, cell activation, and cell division [4]. Inadequate consumption of zinc causes depression and psychosis, stunted growth and development, and immune system damage [5]. Iron inadequacy impairs brain function, growth, reproductive function, and quality of work life, and is mainly responsible for causing anemia [6]. Children under the age of five and pregnant or nursing women have been found to be more vulnerable to deficiencies mainly

caused by iron and zinc in underdeveloped nations [7,8]. The alleviation of micronutrient deficiencies is achieved by a variety of approaches, such as food fortification, supplementation and dietary diversification, but the most effective, affordable, and sustainable way is crop biofortification where crop cultivars improved with nutrition are developed through breeding approaches [9,10].

Phytic acid (PA) (myo-inositol-1,2,3,4,5,6-hexakisphospate) is the principal form of phosphorus (P) storage in seeds [11,12]. During seed maturation, it accumulates as phytate salts, chelating various minerals thereby reducing their bioavailability in monogastric animals [13]. Only ruminants can break down PA among animals that consume seeds due to the presence of phytase producing bacteria in their digestive tracts [12,13]. The PA is mostly deposited in the embryo and scutellum of maize [14]. Monogastric species, including humans, have nearly minimal phytase activity in their digestive tracts; therefore only 10% of the phytate in their meal is broken down with the other 90% being excreted [13,15,16]. Hence, the farmers rearing pigs, poultry, fish, and other monogastric animals must supply supplemental feed containing mineral cations and phosphorus [13]. Moreover, a significant environmental issue known as groundwater eutrophication is caused by the high quantity of excreted P produced from PA [17]. Furthermore, in the digestive system, PA interacts with basic amino acids, seed proteins, and enzymes to produce complexes that may decrease the availability of amino acids, digestibility of proteins, and activity of digestive enzymes [18–20]. However, dietary PA may potentially have health-promoting effects as an antioxidant, anti-cancer agent, or inhibitor of kidney stone development [21].

Maize is one of the most potential staple foods that can improve global food security, as it can be grown in a range of habitats [22]. The livelihood of millions of people worldwide depends on maize due to its critical role in food production, animal feed, and a wide range of industrial applications [23,24]. Nevertheless, maize grains contain more PA than other cereal grains, which considerably lowers the bioavailability of iron and zinc [25,26]. The reduction of PA in maize grains helps in enhancing the bioavailability of iron and zinc in the human gut [12]. Of the various mutants available in maize, *low phytic acid-1* (*lpa1*) possesses great promise for the development of low PA maize hybrids [27]. *Lpa1* affects the capacity of multidrug resistance-associated protein (MRP) transporters to transport and store PA into the vacuole [13]. The mutant *lpa1* allele in homozygous conditions exhibits a 66% reduction in PA content, thereby enhances the bioavailability of positively charged minerals like iron and zinc [27–30]. So far, various studies used temperate germplasm for analysis of the *lpa1* gene [12]. However, the extent of variability in the *Lpa1* gene among the subtropical germplasm is yet to be analyzed. The current work will be useful in understanding the phylogenetic and evolutionary relationship among maize and its orthologues for *lpa1-1*; gene-based *InDel* markers developed can be used to characterize the unknown genotypes for *lpa1* gene; and also supports *lpa1*-based breeding initiatives in maize. Thus, the present study was undertaken to (i) sequence characterize the *Lpa1* gene in a set of mutants and subtropically-adapted wild-type maize inbreds; (ii) discover haplotypes of *Lpa1* gene across diverse subtropical maize inbreds using gene-based *InDel* markers; and (iii) compare the *Lpa1* gene of maize with its orthologues at nucleotide and protein levels.

## 2. Materials and Methods

### 2.1. Genetic Materials

Three low phytic acid mutant inbreds (PMI-LP1-124, A619 *lpa1-1*, and A632 *lpa1-1*) and seven elite wild-type maize genotypes (PMI-PV5, PMI-PV6, PMI-PV7, PMI-PV8, PMI-Q1, PMI-Q2, and PMI-Q3) were chosen for characterization of the full-length *Lpa1* gene at nucleotide and protein levels (Table 1). The *Lpa1* represents the wild-type allele and *lpa1-1* represents the mutant-type allele throughout the text. The wild-type inbreds had a high level of PA and low inorganic phosphorus, whereas the mutant-type genotypes have low phytic acid and high inorganic phosphorus (iP) [27,30]. A diverse panel of 48 genotypes (42 wild-type and six mutant-type inbreds) was used for evaluating the gene-based diversity, and haplotype study (Table S1).

**Table 1.** Details of diverse maize inbreds used in the study for characterization of *Lpa1* gene.

| S. No. | Inbred | Code | Type | Source |
|---|---|---|---|---|
| 1 | PMI-PV5 | *Lpa1*-wild-1 | Wild-type | ICAR-IARI, New Delhi |
| 2 | PMI-PV6 | *Lpa1*-wild-2 | Wild-type | ICAR-IARI, New Delhi |
| 3 | PMI-PV7 | *Lpa1*-wild-3 | Wild-type | ICAR-IARI, New Delhi |
| 4 | PMI-PV8 | *Lpa1*-wild-4 | Wild-type | ICAR-IARI, New Delhi |
| 5 | PMI-Q1 | *Lpa1*-wild-5 | Wild-type | ICAR-IARI, New Delhi |
| 6 | PMI-Q2 | *Lpa1*-wild-6 | Wild-type | ICAR-IARI, New Delhi |
| 7 | PMI-Q3 | *Lpa1*-wild-7 | Wild-type | ICAR-IARI, New Delhi |
| 8 | PMI-LP1-124 | *lpa1-1*-mutant-1 | Mutant-type | ICAR-IARI, New Delhi |
| 9 | A619 *lpa1-1* | *lpa1-1*-mutant-2 | Mutant-type | USDA-ARS, Aberdeen, ID, USA |
| 10 | A632 *lpa1-1* | *lpa1-1*-mutant-3 | Mutant-type | USDA-ARS, Aberdeen, ID, USA |

*2.2. Genomic DNA Isolation, PCR Amplification and Sequencing of Lpa1*

The sodium dodecyl sulfate (SDS) extraction procedure [31] was followed to extract the genomic DNA from the seeds of selected inbreds. The 7292 bp region from the B73 reference genome of the *Lpa1* gene sequence (Gene ID: Zm00001eb003490) from the Zm-B73-REFERENCE-NAM-5.0 version available in Maize Genetics and Genomics Database (maizeGDB), was retrieved. Primer3web v4.1.0 online tool was used to design 13 overlapping primer pairs and were synthesized from Sigma Aldrich Chemical Pvt. Ltd., Bengaluru, India (Table S2). These primers covered the full-length *Lpa1* gene sequence and the product size ranged from 300 to 800 bp. The amplified products were synthesized using a thermal cycler in a 50 µL reaction containing a DNA template, OnePCR$^{TM}$ Mix (GeneDireX Ready to use PCR master mix), and 0.5 µM of each forward and reverse primer. The PCR protocol was carried out with the following PCR conditions using a BIO-RAD model T100$^{TM}$ thermal cycler (Bio-Rad Laboratories Inc., Gurugram, India): (i) initial denaturation for five minutes at 95 °C; (ii) 35 cycles of denaturation at 95 °C, annealing at 55–60 °C (depending on $T_m$ of primer pair), and primer extension at 72 °C for 45 s each step; and (iii) final extension at 72 °C for 10 min. Each PCR reaction was carried out twice and the integrity of each amplicon was checked on a 2.0% Seakem LE agarose gel using 5 µL volume, and the remaining volume was processed for sequencing by Bionivid Technology Private Limited, Bengaluru, India.

*2.3. Alignment of Sequences and Functional Analysis of Lpa1 Gene*

To analyse the raw sequence data, the sequencing results of each fragment were compared to the B73 reference sequence. By arranging 13 overlapping fragments encompassing the entire gene, the complete gene sequence for each genotype was obtained using the Bio-Edit software version 7.0.5.3 [32]. The complete gene sequences of all 10 genotypes and the B73 reference sequence were aligned in MEGA v11(Molecular Evolutionary Genetics Analysis version 11) software using ClustalW and MUSCLE algorithm for identification of SNPs and *InDel* variants among wild-type and mutant inbreds [33]. In order to count the SNPs, *InDels*, polymorphic sites, total mutations, haplotypes, haplotype gene diversity, nucleotide diversity, and *InDel* events, the MEGA alignment file was analysed using DnaSP6 v6.12.03 programme.

*2.4. Gene-Based Diversity Analysis of Lpa1 among the Diverse Maize Genotypes*

Based on the complete *Lpa1* gene sequence among 10 genotypes (seven wild and three mutant-types) and polymorphic regions, 15 gene-based *InDel* markers covering the full length of *Lpa1* were designed (Figure 1; Table S3). The PCR reactions for the *InDel* markers were carried out in a BIO-RAD T100$^{TM}$ thermal cycler (Bio-Rad Laboratories Inc.), all the PCR reactions were performed under the following conditions: initial denaturation at

95 °C for 5 min, 35 cycles of primer annealing at 58–63 °C (depending upon $T_m$) for 45 s, primer extension at 72 °C for 1 min, and final extension at 72 °C for 5 min. The diverse panel of 48 maize inbreds was genotyped using all 15 *InDel* markers. On the basis of *InDel* size, PCR products were run through 4% metaphor agarose and 8% polyacrylamide gel electrophoresis (PAGE). DARwin v6.0 was used to analyze the marker data obtained through the gel profiles, established genetic dissimilarity based on Jaccard's coefficient, and deduced the dendrogram using the Neighbour-joining method [34]. Gene diversity, total allele count, major allele frequency, heterozygosity and the polymorphism information content (PIC) were analysed using PowerMarker v3 [35]. Further, the Newick output file from DARwin software was used to construct a dendrogram through an online server called iTOL (Interactive Tree of Life) [36].

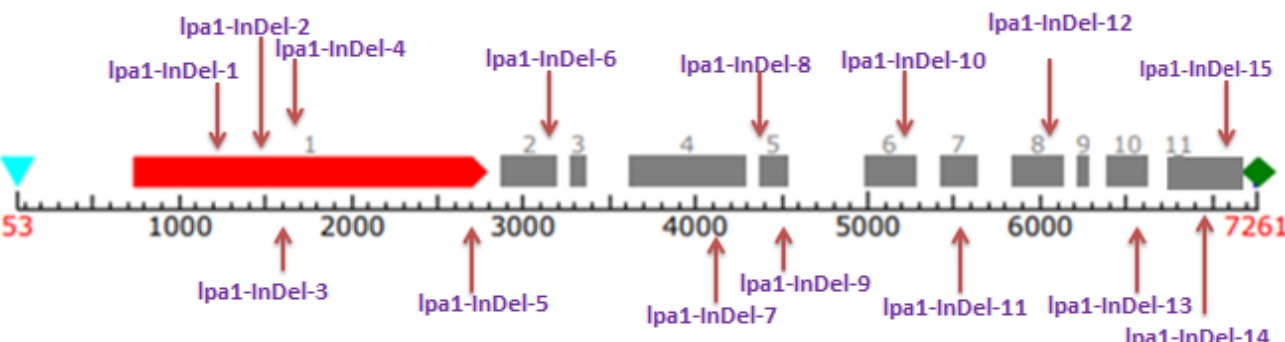

**Figure 1.** Diagrammatic representation of positions of *InDel*-based markers position in *Lpa1* gene (cyan triangle represents TSS (transcription start site); rhombus in green colour represents polyA site, red colour represents first major exon and grey colour represents exons).

### 2.5. Retrieval of Gene and Protein Sequences of Lpa1 Orthologues

On the basis of *Lpa1* sequence similarity (>50%), 26 orthologues were selected for phylogenetic analysis. Nucleotide and protein sequences were retrieved for different accessions of orthologue species. *Lpa1*-wild-Zm00001eb003490 of B73 reference of *Zea mays* was used to compare 10 maize *Lpa1* sequences from the current study, along with 26 orthologues, viz. *Aegilops tauschii* (1), *Ananas comosus* (1), *Asparagus officinalis* (1), *Brachypodium distachyon* (1), *Dioscorea rotundata* (1), *Eragrostis curvula* (1), *E. tef* (1), *Hordeum vulgare* (1), *Leersia perrieri* (1), *Musa acuminata* (2), *Oryza barthii* (1), *O. brachyantha* (1), *O. sativa Japonica Group* (1), *O. sativa Indica Group* (1), *O. rufipogon* (1), *O.nivara* (1), *Panicum hallii HAL2* (1), *Secale cereal* (1), *Setaria viridis* (1), *S. italica*(1), *Sorghum bicolor* (1), *Triticum aestivum* (1), *T. dicoccoides* (1), *T. turgidum* (1), *T. urartu* (1). These were acquired using the BLASTp search tool using an expectation value (e-value) of $\leq 1e^{-5}$ from the publicly accessible Ensembl Plants database [37] (Table 2).

### 2.6. Gene Prediction and Phylogenetic Tree

The maize *Lpa1* gene sequences of 10 inbreds undertaken in the study along with B73 reference sequence and 26 orthologue accessions were subjected to Hidden Markov Model (HMM)-based gene prediction tool, called FGENESH to examine the 5′-UTR, transcription start site, intron-exon boundaries, and poly-A tail [38]. Nucleotide and protein sequences of 37 *Lpa1* accessions including 10 maize sequences from the current study (seven *Lpa1*-wild and three *lpa1-1*-mutant), *Lpa1* reference sequence and 26 orthologue accessions of related crop species (Table 2) were taken for phylogenetic analysis using the CLUSTALW tool in MEGA v11 with 10,000 bootstraps.

**Table 2.** List of maize genotypes and orthologue accessions of *lpa1-1* gene with their gene and protein IDs.

| S. No. | Accessions | | Gene ID | Protein ID |
|--------|------------|---|---------|------------|
| 1 | *Lpa1*-wild-1 | | | |
| 2 | *Lpa1*-wild-2 | | | |
| 3 | *Lpa1*-wild-3 | | | |
| 4 | *Lpa1*-wild-4 | | | |
| 5 | *Lpa1*-wild-5 | *Zea mays* | Nucleotide sequence generated in the present study | Protein sequence translated from nucleotide sequence |
| 6 | *Lpa1*-wild-6 | | | |
| 7 | *Lpa1*-wild-7 | | | |
| 8 | *lpa1-1*-mutant-1 | | | |
| 9 | *lpa1-1*-mutant-2 | | | |
| 10 | *lpa1-1*-mutant-3 | | | |
| 11 | *ZmMRP4*-B73-Ref | | Zm00001eb003490 | A7KVC2 |
| 12 | *Aegilops tauschi* | | AET4Gv20803900 | M8CWG8 |
| 13 | *Ananas comosus* | | Aco010163 | A0A199URG9 |
| 14 | *Asparagus officinalis* | | A4U43_C10F18740 | A0A5P1E459 |
| 15 | *Brachypodium distachyon* | | BRADI_1g75590v3 | I1H9W0 |
| 16 | *Dioscorea rotundata* | | DRNTG_05198 | DRNTG_05198.1 |
| 17 | *Eragrostis curvula* | | EJB05_08061 | TVU48425 |
| 18 | *Hordeum vulgare* | | HORVU.MOREX.r3.4HG0412040 | A0A287PXI6 |
| 19 | *Leersia perrieri* | | LPERR03G03070 | A0A0D9VPF0 |
| 20 | *Musa acuminata* | | Ma08_g12530 | Ma08_t12530.1 |
| 21 | *Musa acuminata* | | Ma11_g02290 | Ma11_t02290.2 |
| 22 | *Oryza barthii* | | OBART03G03330 | A0A0D3FDN4 |
| 23 | *Oryza brachyantha* | | OB03G13350 | J3LJV9 |
| 24 | *Panicum hallii HAL2* | | GQ55_9G618800 | A0A2T7CHT9 |
| 25 | *Secale cereale* | | SECCE5Rv1G0369730 | SECCE5Rv1G0369730.1 |
| 26 | *Setaria viridis* | | SEVIR_9G548400v2 | A0A4U6TCN8 |
| 27 | *Sorghum bicolor* | | SORBI_3001G508200 | A0A1Z5SBX3 |
| 28 | *Triticum aestivum* | | TraesCS5A02G512500 | A0A1D5YDM9 |
| 29 | *Triticum dicoccoides* | | TRIDC4BG057910 | TRIDC4BG057910.7 |
| 30 | *Triticum turgidum* | | TRITD5Av1G244640 | TRITD5Av1G244640.2 |
| 31 | *Triticum urartu* | | TuG1812G0500005238.01 | M8AP62 |
| 32 | *Oryza sativa Japonica Group* | | OsABCC13 | Q10RX7 |
| 33 | *Oryza sativa Indica Group* | | ABCC13 BGIOSGA011835 | A2XCD4 |
| 34 | *Oryza rufipogon* | | ORUFI03G03050 | A0A0E0NPI1 |
| 35 | *Oryza nivara* | | ONIVA03G03060 | A0A0E0GGM3 |
| 36 | *Eragrostis tef* | | Et_s2678-1.30-1.path1 | Et_s2678-1.30-1.mrna1 |
| 37 | *Setaria italica* | | SETIT_033885mg | K4A4T1 |

*2.7. Structural Analysis and Physicochemical Properties of LPA1 Protein*

The LPA1 protein sequence was retrieved from UniProt with the protein ID: A7KVC2. The protein sequences were aligned using the MEGA v11 tool and analyzed for the presence

of point mutations, and conserved regions. The prediction of transition-transversion bias, amino acid composition, and Tajima's Neutrality test were done using the MEGA v11. The chemical and physical properties of proteins, such as the composition of amino acids, molecular weight, isoelectric point (pI), negatively and positively charged amino acids, aliphatic index, instability index and grand average of hydropathicity (GRAVY), were searched using ProtParam online tool (at ExPASy). Self-Optimized Prediction Method from Alignment (SOPMA) was used to study the secondary structure of the LPA1 protein. The functional motifs and domains among the maize and its orthologues protein sequences were identified with the help of the MOTIF search tool (https://www.genome.jp/tools/motif/; accessed on 26 March 2023).

### 2.8. Homology Modeling of LPA1 Protein

SWISS-MODEL was used to generate LPA1 protein models based on homology at various levels of complexity [39]. The best selected homology model was studied with regard to several criteria, including GMQE score >0.5, identity score >50%, and QMEAN score >0.7. The chosen model's PDB file was used for the evaluation of the Ramachandran Plot using the Swiss model. Protein-ligand binding sites were also predicted through SWISS-MODEL.

## 3. Results

### 3.1. Sequence Characterization of Lpa1 Gene among Selected Maize Inbreds

The overlapping amplicons covering the full-length of the *Lpa1* gene in 10 maize genotypes were sequenced and consecutive sequences were aligned with B73 reference to get a complete gene sequence for each genotype. The *Lpa1* gene sequence comparison among the sequenced genotypes revealed the existence of 607 SNPs and 335 *InDels* with an average *InDel* length of 1.55 bp and diversity *k(i)* of 47.418%. The selected 11 maize sequences including B73 reference showed 0.920 sequence conservation (C). A total of 32 conserved regions with *p*-values between 0.0000 and 0.0417 were also found. The phylogenetic tree of 11 maize inbreds including the B73 reference were divided into two clusters, -A and -B, with a sum of branch length (SBL) of 0.1018. The genotypes were correlated together with 10,000 bootstraps and the percentage of replicate trees varied from 35 to 99% (Figure 2A). The Tajima-Nei method produced an evolutionary distance value between 0.00606 and 0.03546. In total, 11 haplotypes were identified among the sequenced maize genotypes with a gene diversity score of 1.00. The observed value for nucleotide diversity (Pi) was 0.01838, Tajima's D value was −1.52327, and theta (per site) from Eta was 0.02679.

Further, the non-synonymous (Ka)/synonymous (Ks) nucleotide diversity ratio was calculated to determine the significance of the observed polymorphisms at the nucleotide sequences. PMI-PV-6 (*Lpa1*-wild-2) had the lowest Ka/Ks ratio among the 10 inbreds that were selected, followed by PMI-Q3 (*Lpa1*-wild-7) and PMI-PV5 (*Lpa1*-wild-1), indicating that they had the fewest non-synonymous mutations and the least amount of evolutionary divergence. The PMI-Q1 (*Lpa1*-wild-5) possessed the largest Ka/Ks ratio; therefore, inferred to possess more significant diversity, which suggested that a particular allele was under selective pressure. The range of Ka/Ks ratio in the case of wild and mutant-type genotypes varied from 0.670 to 1.604 and 0.698 to 0.792, respectively (Table S4).

### 3.2. Characterization of LPA1 Protein among Selected Maize Inbreds

The alignment of protein sequences of wild-type (*Lpa1*) and mutant-type (*lpa1-1*) genotypes showed 841 variables among the maize inbreds. The neighbor-joining method was used to determine the evolutionary relationship of the LPA1 protein among the mutant and wild type genotypes. The measured SBL was 1.138 (Figure 2B). Two main clusters (-P and -Q) were identified based on phylogenetic analysis; each cluster contained four and seven inbreds, respectively. Cluster-P grouped two mutant genotypes (A619 *lpa1-1* and A632 *lpa1-1*) and two wild-type maize genotypes (PMI-PV7 and PMI-PV8). Cluster-Q was

split into two sub-clusters, first sub-cluster possessed two wild-type maize inbreds (PMI-Q1 and PMI-PV5) and one mutant-type inbred (PMI-LP1-124), and the second sub-cluster possessed all the wild-type genotypes (PMI-Q2, PMI-Q3, PMI-PV6) and B73 reference sequence (Protein ID: A7KVC2), respectively (Figure 2B).

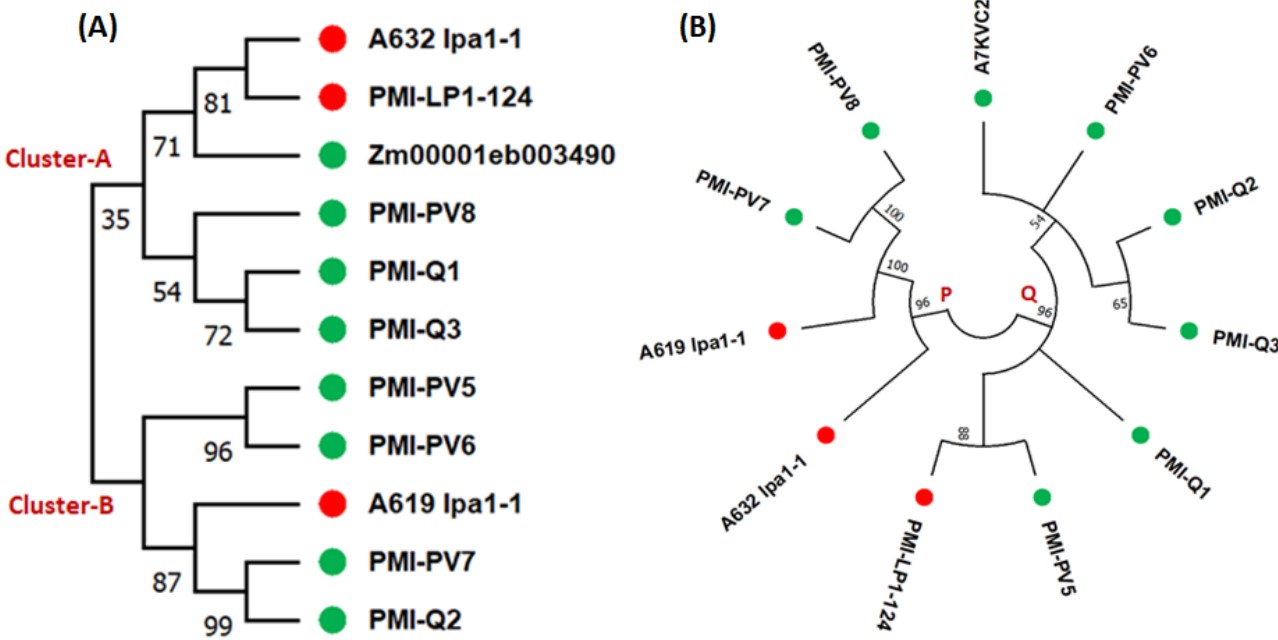

**Figure 2.** Phylogenetic analysis of selected mutant and wild *Lpa1* inbreds on the basis of (**A**) nucleotide sequences (Reference sequence, Gene ID: Zm00001eb003490) and (**B**) protein sequences (Reference sequence, Protein ID: A7KVC2). (Red colour indicates mutant-type inbred and green colour represent wild-type inbred).

### 3.3. Gene-Based Diversity Analysis among Diverse Maize Inbreds Using InDel Markers

Among 335 identified *InDels*, 15 *InDel* sites were exploited to develop markers to study gene-based diversity of *Lpa1* considering the GC-content and necessary distance between two consecutive *InDels* (Table S3). In a panel of 48 diverse-mutant and wild-type maize inbreds, all the 15 *InDel* markers (*lpa1*-InDel-1 to *lpa1*-InDel-15) were polymorphic, producing 1–3 alleles per locus with a total of 38 alleles (Figure S1). The major allele frequency ranged from 0.4583 (*lpa1*-InDel-10) to 0.8750 (*lpa1*-InDel-1) with a mean of 0.6639. PIC among the *InDel* markers ranged from 0.1948 to 0.5423 with an average PIC of 0.3813; 12 markers had PIC of >0.3. The panel of 48 inbreds showed heterozygosity ranging from 0.00 to 0.0417 with an average of 0.0028. Gene diversity varied from 0.219 (*lpa1*-InDel-1) and 0.612 (*lpa1*-InDel-5), with a mean of 0.4521 (Table 3). Using *InDel*-based genotyping of the *Lpa1* gene, 42 haplotypes were formed among 48 inbreds. Of these, six haplotypes were shared among the genotypes viz. Hap6 shared by BML7 and BML45; Hap 16 shared by BK9-2 and HKI1378; Hap17 shared by HKI1040-7 and HKI1344; Hap19 shared by PMI-PV2 and PMI-PV12; Hap27 shared by CML385 and CML410; and Hap31 shared by CML473 and CML474 (Figure 3). The *InDel* markers showed that all the six low phytate maize inbreds with the recessive *lpa1-1* allele had different haplotypes (Figure 3).

**Table 3.** Molecular diversity analyses among the 48 diverse genotypes using gene-based *InDel* markers.

| S. No. | Marker | Major Allele Frequency | No. of Alleles | Gene Diversity | Heterozygosity | PIC |
|---|---|---|---|---|---|---|
| 1 | *lpa1*-InDel-1 | 0.8750 | 2.00 | 0.2188 | 0.0000 | 0.1948 |
| 2 | *lpa1*-InDel-2 | 0.5833 | 3.00 | 0.5729 | 0.0000 | 0.5101 |
| 3 | *lpa1*-InDel-3 | 0.7917 | 2.00 | 0.3299 | 0.0000 | 0.2755 |
| 4 | *lpa1*-InDel-4 | 0.7500 | 2.00 | 0.3750 | 0.0000 | 0.3047 |
| 5 | *lpa1*-InDel-5 | 0.5208 | 3.00 | 0.6120 | 0.0000 | 0.5423 |
| 6 | *lpa1*-InDel-6 | 0.4792 | 3.00 | 0.5720 | 0.0000 | 0.4783 |
| 7 | *lpa1*-InDel-7 | 0.7292 | 2.00 | 0.3950 | 0.0000 | 0.3170 |
| 8 | *lpa1*-InDel-8 | 0.7292 | 2.00 | 0.3950 | 0.0000 | 0.3170 |
| 9 | *lpa1*-InDel-9 | 0.8125 | 2.00 | 0.3047 | 0.0417 | 0.2583 |
| 10 | *lpa1*-InDel-10 | 0.4583 | 3.00 | 0.6007 | 0.0000 | 0.5158 |
| 11 | *lpa1*-InDel-11 | 0.7292 | 3.00 | 0.4210 | 0.0000 | 0.3704 |
| 12 | *lpa1*-InDel-12 | 0.6458 | 3.00 | 0.5026 | 0.0000 | 0.4346 |
| 13 | *lpa1*-InDel-13 | 0.6250 | 3.00 | 0.5078 | 0.0000 | 0.4277 |
| 14 | *lpa1*-InDel-14 | 0.6458 | 2.00 | 0.4575 | 0.0000 | 0.3528 |
| 15 | *lpa1*-InDel-15 | 0.5833 | 3.00 | 0.5174 | 0.0000 | 0.4200 |
| | Mean | 0.6639 | 2.53 | 0.4521 | 0.0028 | 0.3813 |

PIC = Polymorphism information content.

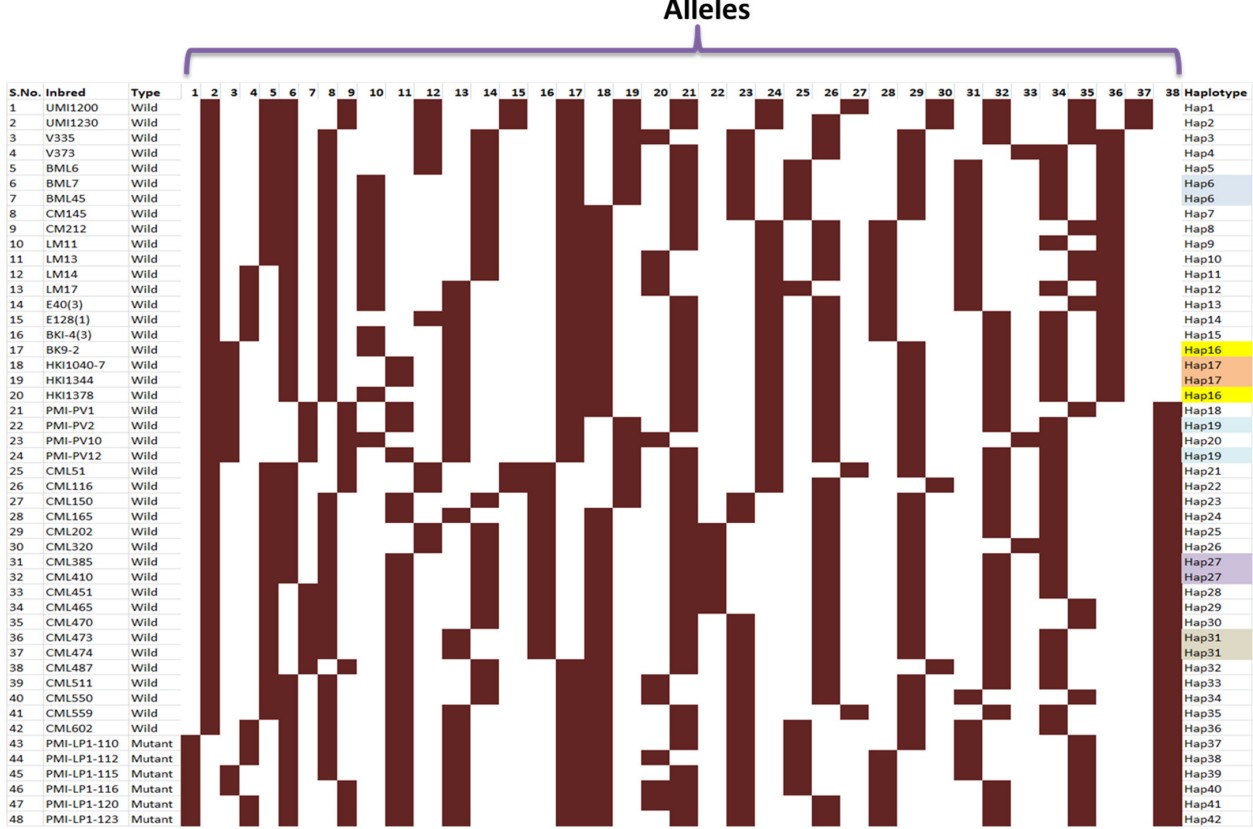

**Figure 3.** Haplotype of *Lpa1* gene using 15 *InDel*-based markers; each row represents the selected mutant and wild genotypes and columns represent the allele for a given marker, black purple colour: presence of DNA band, white box colour: absence of DNA band.

The average genetic dissimilarity was 0.612, which ranged from 0 to 0.9286. The dendrogram divided 48 maize genotypes into two main clusters viz., -R and -S. The 19 inbreds that made cluster-R were further divided into two sub-clusters, with R1 having six mutants and one wild-type genotype with seven haplotypes, and R2 having 12 wild-type genotypes with 10 haplotypes. The 29 maize inbreds that produced cluster-S were divided into three sub-clusters viz., -S1 having four wild-type inbreds with three haplotypes; -S2 having four wild-type genotypes with four haplotypes, and -S3 having 21 wild genotypes with 18 haplotypes (Figure 4).

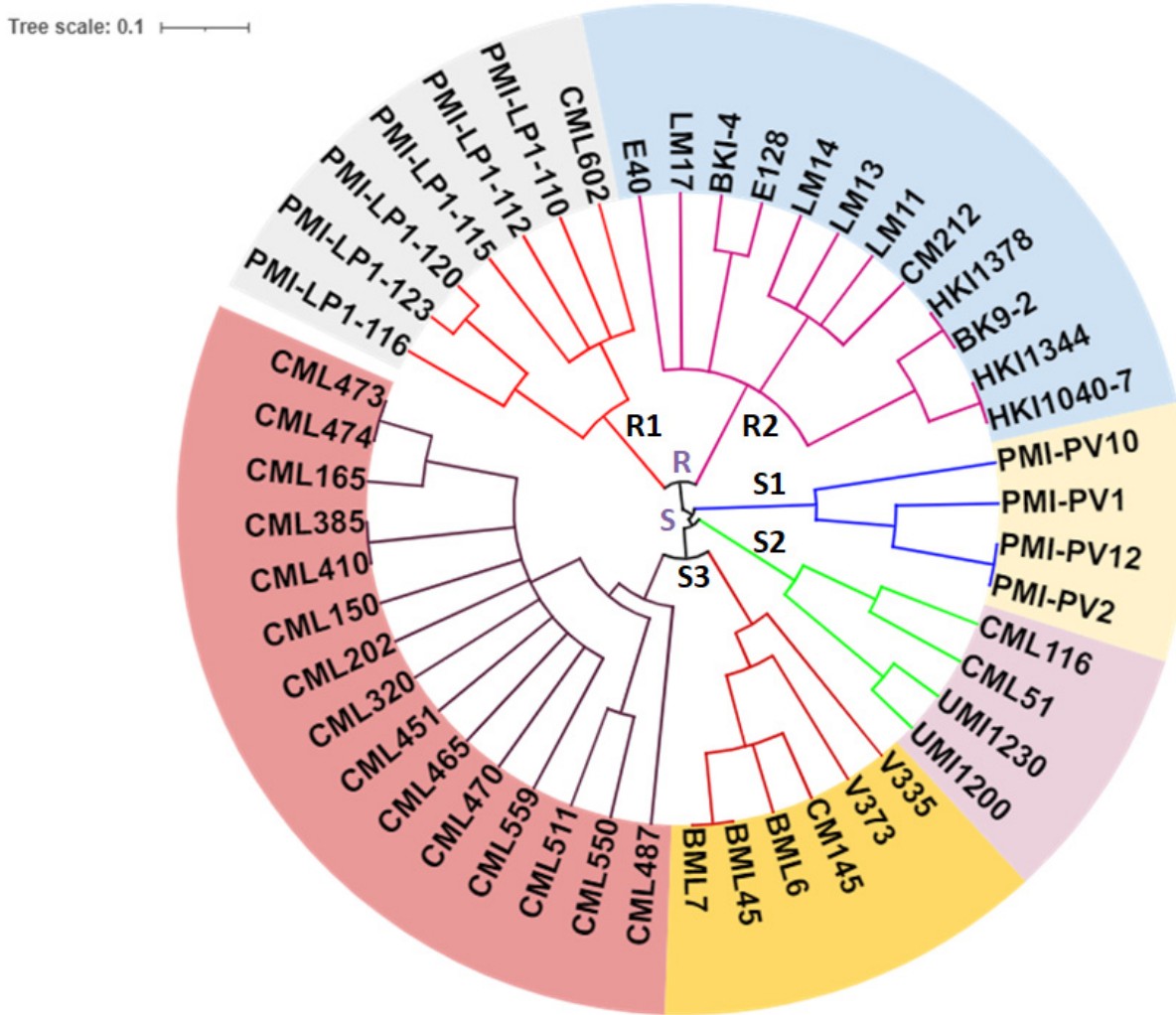

**Figure 4.** Dendrogram representing *lpa1-1* gene-based diversity using *InDel* markers.

### 3.4. Structure of Lpa1 Gene in Maize and Its Orthologues

The TSS of *Lpa1* was located at 53 to 2620 bp upstream in maize genotypes, whereas in orthologues, it was found at 36 to 4774 bp upstream of their respective start site. In maize accessions, the gene coding sequence ranged from 1896 to 4530 bp, whereas in orthologues, it ranged from 387 to 5226 bp. In maize, the number of exons ranged from 11 to 19, while for orthologues, it ranged from 6 to 14 (Table S5). The length of exon ranged in maize accessions from 21 to 2314 bp, and in the sequences of the orthologues, it varied from 6 to 2076 bp. The analysis showed that the length of introns across maize genotypes varied from 36 to 892 bp, whereas in the case of orthologues, it ranged from 55 to 2653 bp. The location of poly-A site among maize inbreds was found from 6726 to 7307 bp downstream of start, the same in orthologues was from 4972 to 13,813 bp.

A total of 37 accessions from 16 different crops were selected to study the evolutionary relationship of the *Lpa1* gene (Table 2). According to the phylogenetic analysis, the evolutionary tree contained branches having an average branch length of 3.85. In the bootstrap test (10,000 replications), the percentage of duplicate trees with the relevant taxa clustered together ranged from 51 to 100%. The phylogenetic tree, based on nucleotide sequence of 37 accessions, had two major clusters viz., -C and -D. The cluster-C was subdivided into two sub-clusters viz., -C1 and -C2. The sub-cluster-C1 had 15 orthologues viz. *T. urartu* (TuG1812G0500005238.01), *T. aestivum* (TraesCS5A02G512500), *T. turgidum* (TRITD5Av1G244640), *T. dicoccoides* (TRIDC4BG057910), *A. tauschii* (AET4Gv20803900), *S. cereale* (SECCE5Rv1G0369730), *H. vulgare* (HORVU.MOREX.r3.4HG0412040), *B. distachyon* (BRADI_1g75590v3), *O. brachyantha* (OB03G13350), *L. perrieri* (LPERR03G03070), *O. sativa var. indica* (ABCC13 BGIOSGA011835), *O. barthii* (OBART03G03330), *O. sativa var. japonica* (OsABCC13), *O. rufipogon* (ORUFI03G03050) and *O. nivara* (ONIVA03G03060). All the 10 maize genotypes including three *lpa1-1* maize mutants (A619 *lpa1-1*, A632 *lpa1-1* and PMI-LP1-124) and B73 reference sequence (Gene ID: Zm00001eb003490) were grouped under sub-cluster-C2. Further, maize genotypes showed a strong association with six orthologues viz. *S. bicolor* (SORBI_3001G508200), *P. hallii* (GQ55_9G618800), *S. italica* (SETIT_033885mg), *S. viridis* (SEVIR_9G548400v2), *E. tef* (Et_s2678-1.30-1.path1) and *E. curvula* (EJB05_08061) in the sub-cluster- C2 (Figure 5). The cluster-D possessed five orthologue accessions viz. *D. rotundata* (DRNTG_05198), *A. officinalis* (A4U43_C10F18740), *A. comosus* (Aco010163), *M. acuminata* (Ma08_g12530), and *M. acuminata* (Ma11_g02290) (Figure 5).

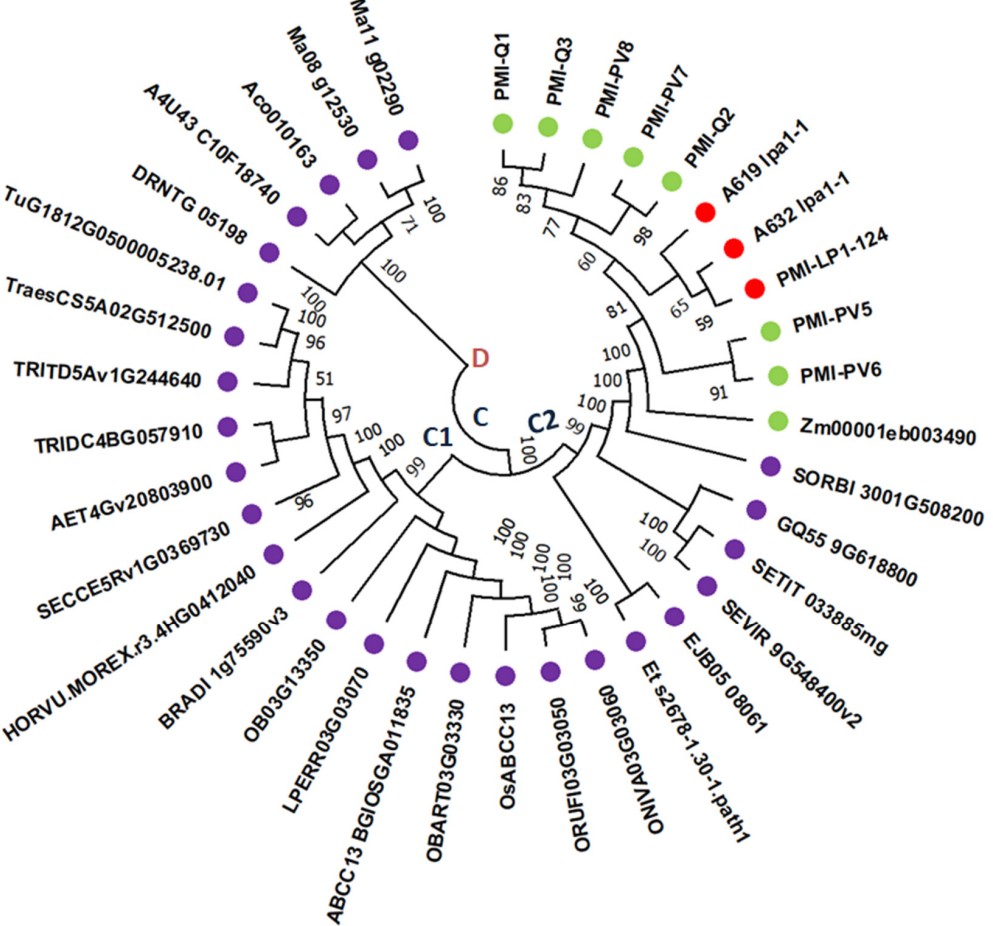

**Figure 5.** Evolutionary relationship of *lpa1-1* gene in maize and its orthologues in selected monocots (Green dots are *Lpa1* wild-type maize inbreds; Red dots are *lpa1-1* mutant-type maize inbreds and purple are orthologues).

### 3.5. Structure of LPA1 Protein in Maize and Its Orthologues

The phylogenetic tree between the 11 maize genotypes and its 26 orthologues was divided into two major clusters called cluster-M and -N; with cluster-M was further divided into sub-clusters -M1, -M2, -M3 -M4 and M5 (Figure S2). All the maize protein sequences including B73 reference along with four orthologues viz. *S. bicolor* (A0A1Z5SBX3), *P. hallii HAL2* (A0A2T7CHT9), *S. viridis* (A0A4U6TCN8) and *S. italica* (K4A4T1) were grouped in sub-cluster-M1 (Figure S2). The M2 sub-cluster as a solo cluster possessed *E. curvula* (TVU48425) orthologue. The sub-cluster-M3 possessed seven orthologues viz. *O. brachyantha* (J3LJV9), *L. perrieri* (A0A0D9VPF0), *O. barthii* (A0A0D3FDN4), *O. sativa var. indica* (A2XCD4), *O. sativa var. japonica* (Q10RX7), *O. rufipogon* (A0A0E0NPI1), and *O. nivara* (A0A0E0GGM3). The *B. distachyon* (I1H9W0) also formed as a solo cluster (sub-cluster-M4). The sub-cluster-M5 possessed seven orthologues viz. *H. vulgare* (A0A287PXI6), *S. cereale* (SECCE5Rv1G0369730.1), *T. urartu* (M8AP62), *T. dicoccoides* (TRIDC4BG057910.7), *A. tauschi* (M8CWG8), *T. aestivum* (A0A1D5YDM9), and *T. turgidum* (TRITD5Av1G244640.2) (Figure S2). On the contrary, another major cluster-N comprised of six orthologues viz. *M. acuminata* (Ma08_t12530.1), *M. acuminata* (Ma11_t02290.2), *D. rotundata* (DRNTG_05198), *A. comosus* (A0A199URG9), *E. tef* (Et_s2678-1.30-1.mrna1) and *A. officinalis* (A0A5P1E459) (Figure S2).

### 3.6. Homology Modeling of LPA1 Protein

The sequence of LPA1 protein was searched for templates in SWISS-Model in order to find a similar pattern. There were 23,418 templates, in all, it matched the target sequence. A heuristic model was used to reduce this list's length to 50. A normalised BLOSUM62 substitution matrix was used to calculate the template sequence similarity. The top four templates were used for modeling with Global Model Quality Estimation (GMQE) scores between 0.54 and 0.57, which accounted for the QMEANDisCo Global score of $0.64 \pm 0.05$ and $0.68 \pm 0.05$. The sequence identity varied from 35.92 to 36.38% among the top four models. All four models [6uy0.1.A (bovine), 6bhu.1.A (bovine), 5uj9.1.A (bovine) and 5uja.1.A (bovine)] were monomeric states and depicted bovine multidrug resistance protein 1 (MRP1). The QMEAN observed for all four models [6uy0.1.A (bovine), 6bhu.1.A (bovine), 5uj9.1.A (bovine) and 5uja.1.A (bovine)] were $-2.86, -3.41, -4.09$ and $-3.65$, respectively. The highest sequence identity was observed in 6uy0.1.A (36.38%). Ramachandran plot depicted that 95.32% of residues lay in favoured regions and only 0.79% were outliers. The MolProbity score was 1.10 and the clash score was 0.90. The protein 6uy0.1.A was a multidrug resistance-associated protein 1 that encoded the ATP binding cassettes and was one of the top comparable templates (Figure S3).

### 3.7. Domains, Motifs and Features of LPA1 Protein

The 52 domains and features were annotated in the LPA1 protein sequence (A7KVC2), which was associated with 549 variants. The major domains were ABC (ATP-binding Cassette sub-family c), P-loop containing nucleoside triphosphate hydrolase, and AAA-ATPases associated with a variety of cellular activities (Table S6). With the help of the MOTIF search tool, 18 motifs were found in the protein sequence of LPA1 (A7KVC2). The 18 motifs were ABC transporter transmembrane region, ABC transporter, RecF/RecN/SMC N terminal domain, 50S ribosome-binding GTPase, P-loop containing region of AAA domain, AAA ATPase domain, AAA domain, putative AbiEii toxin, Type IV TA system, Helicase HerA, central domain, Dynamin family, Methylmalonyl Co-A mutase-associated GTPase MeaB, Type II/IV secretion system protein, Conserved hypothetical ATP binding protein, AAA domain, FtsK/SpoIIIE family, Type IV secretion-system coupling protein DNA-binding domain and Double-GTPase 2. In the case of wild-type genotypes, the LPA1 protein motifs varied from 15 to 20, whereas in case of mutant-type, it varied from 13 to 20 motifs. The least number of motifs were found in PMI-LP1-124 (13 motifs), PMI-PV7 (15 motifs) and PMI-Q1 (16 motifs) among maize protein sequences. The maximum number of motifs (20 motifs) were found in PMI-PV-5, PMI-PV-6 and A632 *lpa1-1* among

maize protein sequences. The additional motifs were NB-ARC domain (PMI-PV5, PMI-PV6 and PMI-Q1, PMI-Q3 and A632 *lpa1-1*), Peptidase family S49 N-terminal (PMI-PV5), DEAD/DEAH box helicase (PMI-PV6 and PMI-Q3), alpha/beta hydrolase fold (PMI-PV7) and citrate transporter (A632 *lpa1-1*).

All the orthologue protein sequences were predicted with major motifs, which were present in the reference sequence of LPA1-1 viz. ABC transporter transmembrane region, ABC transporter, AAA domain, putative AbiEii toxin, Type IV TA system, AAA domain, Helicase HerA, central domain, AAA ATPase domain, Methylmalonyl Co-A mutase-associated GTPase MeaB, FtsK/SpoIIIE family, Double-GTPase 2, conserved hypothetical ATP binding protein. Motifs other than reference sequences were also present in some orthologue protein sequences. NB-ARC domain was present in some maize genotypes as well as in orthologues viz., *E. curvula* (TVU48425), *H. vulgare* (A0A287PXI6), *M. acuminata* (Ma08_t12530.1), *T. turgidum* (TRITD5Av1G244640.2), *E. tef* (Et_s2678-1.30-1.mrna1). Zeta toxin motifs were present in *L. perrieri* (A0A0D9VPF0), *O. brachyantha* (J3LJV9), Coilin N-terminus in *A. comosus* (A0A199URG9), *M. acuminata* (Ma08_t12530.1), *M. acuminata* (Ma11_t02290.2), Magnesium chelatase, subunit ChlI in *A. officinalis* (A0A5P1E459), *E. curvula* (TVU48425), *M. acuminata* (Ma11_t02290.2), *E. tef* (Et_s2678-1.30-1.mrna1), CobW/HypB/UreG, nucleotide-binding domain in *O. barthii* (A0A0D3FDN4), DNA mimic ocr in *O. barthii* (A0A0D3FDN4), *O. sativa Japonica Group* (Q10RX7), *O. sativa Indica Group* (A2XCD4), *O. nivara* (A0A0E0GGM3), DEAD/DEAH box helicase in *D. rotundata* (DRNTG_05198.1), *S. bicolor* (A0A1Z5SBX3), and Pre-mRNA-splicing factor PRP9 N-terminus in *M. acuminata* (Ma08_t12530.1).

### 3.8. Physicochemical Properties of LPA1 Protein in Maize and Selected Orthologues

In maize, 1510 amino acids long LPA1 (A7KVC2) protein had a coding sequence of 4530 bp. The total number of negatively charged residues (Asp + Glu) in reference sequence was 145, whereas the total number of positively charged residues (Arg + Lys) was 155. Leucine was the predominant amino acid that ranged from 10.8 to 13.9% among selected maize protein sequences; followed by the alanine (5.2 to 10.8%). The highest amount of amino acid in selected orthologues was leucine (11.2 to 12.7%), followed by alanine which ranged from 7.2 to 10.8%. However, in the case of *T. turgidum* and *T. urartu*, serine was the second highest amino acid at 8.7% and 8.5%, respectively). At 280 nm, the measured extinction coefficients in water were 214,320 $M^{-1}cm^{-1}$ assuming that all pairs of cysteine residues form cystines and 212,570 $M^{-1}cm^{-1}$ assuming that all pairs of cysteine residues were reduced. GRAVY and the aliphatic index were 0.19 and 107.19, respectively, indicating that the compound was polar (Table 4). The aliphatic index of all maize protein sequences ranged from 96.74 to 109. High aliphatic index values demonstrated thermostable solubility nature. The GRAVY value of the reference protein sequence reflects its polar nature; negative values reflect the non-polar nature of the peptide, whereas positive value reflects its polar nature. All selected maize protein sequences were polar in nature except PMI-PV5, PMI-PV7, PMI-LP1-124 and A619 *lpa1-1*. The reference LPA1 protein was computed as unstable with an instability index of 46.28. The PMI-Q3 had the highest instability index of 50.07 among selected maize genotypes. The secondary structure of the LPA1-1 protein of the reference sequence had 797 alpha helices, 216 extended strands, 84 beta turns and 413 random coils (Figure S3). In the case of orthologues, all the protein sequences were polar in nature due to positive GRAVY value except *O. brachyantha* and *E. tef.* All the selected maize LPA1 protein sequences and their orthologues were unstable, as they all had an instability index of more than 40 (Table 4). The aliphatic index varied from 102.50 to 107.40 among selected orthologue LPA1 protein sequences. All orthologues were highly thermostable and hydrophobic in nature.

**Table 4.** Physico-chemical characteristics of LPA1-1 protein in maize and its selected orthologues protein.

| S. No. | Sequence | Crop Species | Amino Acid | Molecular Weight | Isoelectric Point (pI) | Negatively Charged aa (Asp + Glu) | Positively Charged aa (Arg + Lys) | Instability Index | Aliphatic Index | GRAVY |
|---|---|---|---|---|---|---|---|---|---|---|
| 1 | A7KVC2 | | 1510 | 166,790.19 | 8.44 | 145 | 155 | 46.28 | 107.19 | 0.19 |
| 2 | PMI-PV5 | | 1196 | 131,732.1 | 7.16 | 129 | 128 | 47.47 | 101.41 | −0.006 |
| 5 | PMI-PV6 | | 1127 | 122,784.47 | 8.53 | 111 | 121 | 42.87 | 105.98 | 0.094 |
| 4 | PMI-PV7 | | 1076 | 118,397.43 | 9.3 | 105 | 135 | 48.05 | 102.49 | −0.02 |
| 3 | PMI-PV8 | | 1275 | 140,824.85 | 9.36 | 105 | 145 | 49.87 | 105.86 | 0.174 |
| 7 | PMI-Q1 | *Zea mays* | 1085 | 119,470.42 | 7.87 | 101 | 104 | 47.4 | 108.47 | 0.229 |
| 6 | PMI-Q2 | | 1186 | 130,285.93 | 6.7 | 115 | 111 | 47.28 | 106.69 | 0.173 |
| 8 | PMI-Q3 | | 1176 | 129,741.83 | 8.76 | 117 | 134 | 50.07 | 102.82 | 0.055 |
| 9 | A632 *lpa 1-1* | | 1134 | 124,915.64 | 7.66 | 115 | 117 | 48.58 | 109 | 0.186 |
| 10 | PMI-LP1-124 | | 1004 | 111,797.38 | 8.89 | 98 | 114 | 48.23 | 96.74 | −0.024 |
| 11 | A619 *lpa1-1* | | 632 | 70,300.03 | 6.59 | 76 | 74 | 47.23 | 100.55 | −0.112 |
| 12 | M8CWG8 | *Aegilops tauschi* | 1504 | 166,057.78 | 8.76 | 150 | 166 | 46.28 | 102.51 | 0.096 |
| 13 | A0A199URG9 | *Ananas comosus* | 1522 | 168,424.55 | 8.27 | 152 | 159 | 42.97 | 103.21 | 0.168 |
| 14 | A0A5P1E459 | *Asparagus officinalis* | 583 | 65,439.23 | 8.28 | 58 | 61 | 44.06 | 103.22 | 0.108 |
| 15 | I1H9W0 | *Brachypodium distachyon* | 1505 | 165,978.66 | 8.1 | 151 | 156 | 43.57 | 104.51 | 0.162 |
| 16 | DRNTG_05198.1 | *Dioscorea rotundata* | 1522 | 170,326.47 | 8.36 | 148 | 156 | 40.46 | 106.74 | 0.185 |
| 17 | TVU48425 | *Eragrostis curvula* | 1502 | 165,632.69 | 7.62 | 151 | 153 | 43.09 | 105.88 | 0.185 |
| 18 | A0A287PXI6 | *Hordeum vulgare* | 1080 | 120,082.52 | 8.22 | 115 | 120 | 43.63 | 104.93 | 0.096 |
| 19 | A0A0D9VPF0 | *Leersia perrieri* | 1505 | 166,391.42 | 7.23 | 154 | 154 | 44.29 | 106.35 | 0.177 |
| 20 | Ma08_t12530.1 | *Musa acuminata* | 1511 | 168,530.23 | 8.18 | 152 | 158 | 42.51 | 107.4 | 0.184 |
| 21 | Ma11_t02290.2 | | 1502 | 168,023.89 | 8.56 | 151 | 163 | 43.58 | 106.76 | 0.183 |
| 22 | A0A0D3FDN4 | *Oryza barthii* | 1385 | 154,216.68 | 6.28 | 152 | 144 | 45.36 | 103.34 | 0.089 |
| 23 | J3LJV9 | *Oryza brachyantha* | 1128 | 125,411.6 | 5.88 | 135 | 121 | 44.12 | 102.15 | −0.015 |
| 24 | A0A2T7CHT9 | *Panicum hallii HAL2* | 1504 | 165,872.84 | 8.19 | 146 | 152 | 44.38 | 106.8 | 0.178 |
| 25 | SECCE5Rv1G0369730.1 | *Secale cereale* | 1509 | 165,870.61 | 7.97 | 152 | 156 | 43.63 | 105.15 | 0.173 |
| 26 | A0A4U6TCN8 | *Setaria viridis* | 1507 | 166,595.77 | 8.48 | 145 | 155 | 45.28 | 106.77 | 0.177 |

**Table 4.** *Cont.*

| S. No. | Sequence | Crop Species | Amino Acid | Molecular Weight | Isoelectric Point (pI) | Negatively Charged aa (Asp + Glu) | Positively Charged aa (Arg + Lys) | Instability Index | Aliphatic Index | GRAVY |
|---|---|---|---|---|---|---|---|---|---|---|
| 27 | A0A1Z5SBX3 | *Sorghum bicolor* | 1512 | 166,559.92 | 8.31 | 146 | 154 | 44.67 | 107.38 | 0.203 |
| 28 | A0A1D5YDM9 | *Triticum aestivum* | 1441 | 159,222.76 | 7.06 | 150 | 149 | 42.19 | 104.74 | 0.155 |
| 29 | TRIDC4BG057910.7 | *Triticum dicoccoides* | 1483 | 163,430.9 | 8.09 | 150 | 155 | 42.55 | 105.14 | 0.161 |
| 30 | TRITD5Av1G244640.2 | *Triticum turgidum* | 1080 | 119,946.28 | 8.1 | 115 | 119 | 45.13 | 104.57 | 0.091 |
| 31 | M8AP62 | *Triticum urartu* | 1346 | 149,368.05 | 6.57 | 145 | 141 | 41.59 | 103.4 | 0.1 |
| 32 | Q10RX7 | *Oryza sativa Japonica Group* | 1446 | 159,766.09 | 7.78 | 153 | 155 | 43.48 | 102.95 | 0.085 |
| 33 | A2XCD4 | *Oryza sativa Indica Group* | 1357 | 149,630.27 | 6.39 | 148 | 141 | 44.49 | 104.52 | 0.11 |
| 34 | A0A0E0NPI1 | *Oryza rufipogon* | 1448 | 160,012.38 | 7.47 | 154 | 155 | 43.74 | 102.81 | 0.084 |
| 35 | A0A0E0GGM3 | *Oryza nivara* | 1411 | 155,399.84 | 7.28 | 149 | 149 | 44.42 | 102.54 | 0.075 |
| 36 | Et_s2678-1.30-1.mrna1 | *Eragrostis tef* | 711 | 79,605.95 | 5.09 | 94 | 76 | 47.16 | 105.88 | −0.005 |
| 37 | K4A4T1 | *Setaria italica* | 1435 | 158,725.42 | 8.53 | 142 | 153 | 45.2 | 105.74 | 0.136 |

## 4. Discussion

Large populations worldwide suffer from micronutrient malnutrition, inflicting severe socio-economic losses which creates a global concern [40]. The deficiency of iron and zinc can be effectively addressed by developing low PA maize cultivars possessing mutant *lpa1-1* allele [41,42]. The *Lpa1* mutation was first reported by Dr. Victor Raboy from USDA, with the help of chemical mutagenesis using ethyl methanesulfonate (EMS) [43]. In the *lpa1-1* mutant, PA is not deposited within the protein-storage vacuoles leading to the feedback inhibition of the synthesis of PA [44]. In the present investigation, we characterized the *Lpa1* sequence in diverse maize inbreds to examine the allelic variations, gene-based diversity, and phylogenetic relationship of the *Lpa1* gene in maize and its orthologues, for enhancing opportunities of lowering PA, and hence improving the nutritional seed quality in cereal-based crops.

### 4.1. Allelic Variations in Lpa1 Gene in Maize Genotypes

The development of desirable crop cultivars depends on utilizing suitable allelic variants of a specific trait [45,46]. The SNP based on the C (wild) to T (mutant) transition at amino acid position 1432 of the 10th exon in the *Lpa1* gene [44], two dominant markers were developed by Abhijith et al. [30]. These markers were later utilized in a molecular breeding programme to select the genotypes with low phytic acid in maize [47]. Based on this causal polymorphic SNP (C to T), seven wild-type and three mutant-type inbreds were selected for full-length *Lpa1* gene sequence characterization in the present study. The complete gene sequence analysis revealed adequate variation in both wild-type (*Lpa1*) and mutant (*lpa1-1*) alleles across diverse maize inbreds. The nucleotide diversity of wild-type inbreds was higher than that of mutants, due to their diverse ancestors [48,49]. High nucleotide diversity was confirmed by the Tajima's neutrality (D) test value, which was shown to be significant in a negative direction and indicated that the pressure on selection was caused by reduced average heterozygosity across the genotypes [50,51]. The clustering pattern of nucleotide sequences indicated that different variations exist between mutant- and wild-type sequences of the *Lpa1* gene. The reference sequence showed more similarity with mutants viz., PMI-LP1-124 and A632 *lpa1-1*. The evolutionary distance is regarded to be a powerful predictor of the neutral mode of evolution based on synonymous (Ks) and nonsynonymous (Ka) sites [52]. The direction and strength of natural selection operating on protein-coding genes is determined from the Ka/Ks ratio and a ratio of <1 denotes stabilizing selection [53]. All of the sequenced maize genotypes in the present study had a Ka/Ks ratio of <1, with the exception of *lpa1*-wild-5 (PMI-Q1).

Gene-based markers used in genetic diversity analysis provide opportunity to evaluate genetic relatedness and locus conservation within genes with greater efficiency [52]. In comparison to Hossain et al. [54] (1.81 alleles/loci), Chhabra et al. [49] (2.00 alleles/loci) and Katral et al. [52] (2.27 alleles/loci), the average number of alleles per locus detected in the present investigation was found to be higher (2.53 alleles/loci). The present research reported lower mean heterozygosity, which showed that most of the loci were homozygous and the alleles were stable [52]. Of the 48 genotypes examined in the study formed 42 haplotypes. Chhabra et al. [49] identified 44 haplotypes, Katral et al. [52] identified 41 haplotypes and Chhabra et al. [55] identified 47 haplotypes of 48 maize genotypes based on *su1*, *Zmfatb* and *sh2* gene-based markers, respectively and Shin et al. [56] identified 14 haplotypes of 15 sweet corn accessions using SNP based markers. Sequencing technologies has made it simpler to generate haplotype information, which aids in determining ancestry and demographic history [57]. The *InDel* markers discovered in this study offer a great promise to accurately and economically identify haplotypes in unknown germplasm. The *InDel*-based markers offer several advantages over polymorphic markers such as SNP, which includes excellent precision and high stability that help in resolving any ambiguity that may develop during genetic study [58]. These *InDel* markers from the current study can be used in marker-assisted selection (MAS) if polymorphic between recipient and donor parents.

There were 11 different haplotypes found when the *Lpa1* gene was sequence character-ized in distinct mutant and wild genotypes. Further, 11 maize genotypes were shown to be distinct from one another, and the cluster analysis validated this finding. *S. bicolor*, *S. italica*, *S. viridis* and *P. hallii* showed very close association with *Lpa1* reference sequence based on nucleotide phylogenetic tree. *P. hallii* is closely linked to plants in the Andropogoneae fam-ily, including sorghum and maize [59]. Around 27 million years ago, *Setaria sp.* split off from *Sorghum* and *Zea mays*, and the same was seen in the new sub-group [49]. The *MRP5* gene of *Arabidopsis thaliana* and *MRP4* gene of maize coded by *Lpa1* were closely related [60].

### 4.2. Variation in LPA1 Protein among Maize Genotypes

The mutant *lpa1-1* was due to ethyl methanesulfonate, carried an alanine (wild-type) to valine (mutant-type) point mutation from C (wild) to T (mutant) SNP at 1432 amino acid po-sition in LPA1 protein [43,44]. The amino acid (alanine) at this position is conserved in MRP protein and located in the second ATP-binding domain of wild-type genotypes, whereas mutant genotypes possessed truncated MRP protein with valine at 1432 position with other amino acids unchanged [44]. The maize *lpa1-1* mutant lacks the MRP-ABC transporter that mainly expresses in embryos as well as in immature endosperm, germinating seed, and vegetative tissues [44]. The compartmentalization and transfer of PA is affected by the ABC transporter. It plays a role in the elimination of PA from the cytosol or the trafficking of vesicles, either directly or indirectly. The reference protein sequence showed strong a corre-lation with the wild-type protein sequence (PMI-PV6) in the protein-based phylogenetic tree. In phylogenetic analysis, all three mutant protein sequences (PMI-LP1-124, A619 *lpa1-1*, and A632 *lpa1-1*) formed a sub-group. One wild-type protein sequence (PMI-PV5) also belonged to this sub-group and showed a strong relationship with the A632 *lpa1-1* protein sequence. The protein sequences from orthologues like *S. bicolor*, *P. hallii*, *S. viridis*, and *S. italica* indicated a close connection with maize protein sequences. The physico-chemical characteristics of the LPA1 protein in maize and its orthologues showed that *D. rotundata* amino acid residues and molecular weight were the highest when compared to those of other orthologues. This is because it has longer coding sequences and more exons [61]. Protein thermostability is indicated by the aliphatic index; the greater the index, the higher the thermostability [62]. All the maize sequences and selected orthologue sequences had a more aliphatic index, it suggested that all selected maize and orthologue proteins were more thermostable and could withstand greater temperatures [61,62]. Positive numbers on the GRAVY scale denote peptide hydrophobicity, whereas negative values show a protein hydrophilic nature [52]. Every protein in the current study had a GRAVY score between $-0.112$ and $0.229$, indicating that it is hydrophobic in nature and they have protection from the water except PMI-PV5 ($-0.006$), PMI-PV7 ($-0.02$), PMI-LP1-124 ($-0.024$), A619 *lpa1-1* ($-0.112$), *O. brachyantha* ($-0.015$) and *E. tef* ($-0.005$). The development of three-dimensional models enables a better comprehension of the structural variations of the proteins because the general structure of these enzymes appears to be preserved [63]. In order to fully un-derstand the structural variations between the wild-type (Reference sequence, Protein ID: A7KVC2) and mutant-type (A632 *lpa1-1*) of maize protein sequence, homology modeling of the LPA1 protein was performed. The template of 8f4b.1.A showed more sequence identity with mutant-type (A632 *lpa1-1*) protein (Figure 6), whereas template 6uy0.1.A revealed more sequence identity with the reference sequence (Protein ID: A7KVC2). It was found in the second ATP-binding domain of MRP proteins. It was highly likely that this amino acid mutation caused the phenotype of the *lpa1-1* allele [44]. Most of the selected orthologues and maize protein sequences had higher isoelectric points (pI: >7), indicating that they were likely to precipitate in simple buffers and exhibit highly conserved activities [49].

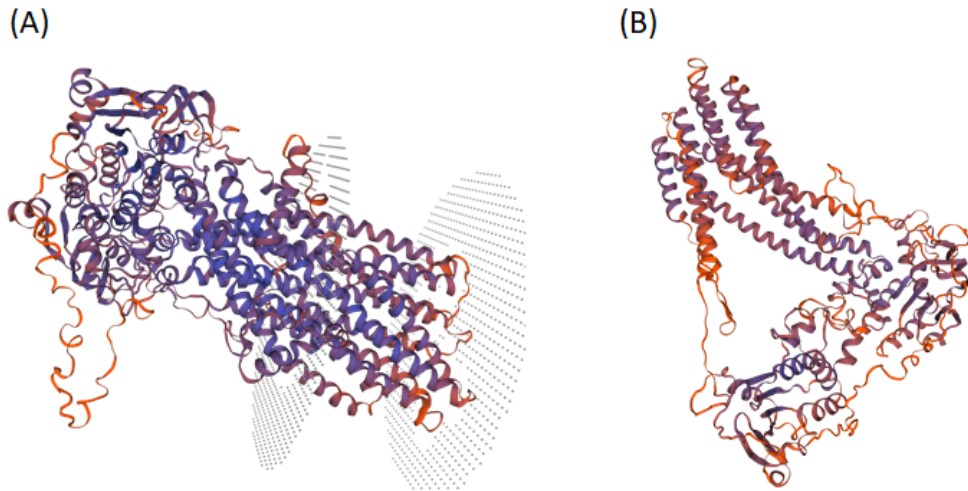

**Figure 6.** Homology models of (**A**) Reference protein (protein ID: A7KVC2); (**B**) Mutant-type [A632 *lpa1-1*] protein obtained through Swiss Model.

## 5. Conclusions

The present study showed that the link between several orthologues and their development from a common ancestor can be better understood through phylogenetic analysis. It was established that the phylogenetic tree derived from nucleotide and protein sequence data was reliable for cluster formation. Thus, the current study demonstrated the existence of allelic variation in the *Lpa1-1* gene in maize, with as many as 42 haplotypes among the diverse panel of 48 inbreds. The gene-based *InDel* markers developed in the present study can be used to characterize unknown genotypes for the *Lpa1* gene. The study also revealed two major domains viz., ABC transporter transmembrane region and ABC transporter in LPA1 protein. All the chosen accessions of maize and other monocots possessed the conserved functional domain. The current study also provided information on the LPA protein in maize and its orthologues and also their physicochemical features. Exon and intron length variation in the *Lpa1* gene revealed relationships in the evolution of maize and its orthologues. Phylogenetic study based on nucleotide as well as protein, maize *Lpa1* sequence is more related to *S. bicolor*, *P. hallii*, *S. virdis* and *S. italica* than other selected poaceae family orthologues. The information generated here will be useful in determining the degree of genetic variation and accelerating the *lpa1*-based breeding programme in maize. This is the first report on a thorough investigation of the *Lpa1* gene and its protein in maize and related orthologues.

**Supplementary Materials:** The following supporting information can be downloaded at: https://www.mdpi.com/article/10.3390/agriculture13071286/s1, Figure S1: Representative gel image of PCR profile of marker *lpa1*-InDel-14F/R; Figure S2: Protein based phylogenetic tree of maize and its orthologues (wild-type maize genotypes indicated with green dot labels whereas red dot represented the mutant-type genotypes of maize, and orthologues represented by purple dots); Figure S3: Homology based LPA1 protein (protein ID: A7KVC2), (a) alignment with 6uy0.1.A (Bovine) template by Swiss Model; (b) Local quality estimation (c) Ramachandran plot using Swiss Model (d) Secondary structure by SOPMA (e) amino acid compositions by predict protein tool; Table S1: Information on the 48 maize inbreds utilised in an *InDel*-based gene diversity analysis (WT: wild-type; MT: mutant-type); Table S2: Details of overlapping primers developed for sequencing full length *Lpa1-1* gene; Table S3: Primer details employed for gene-based diversity in *Lpa1-1* gene; Table S4: Evolutionary distance and synonymous and non-synonymous scores in comparison with reference *Lpa1* (Gene ID: Zm00001eb003490); Table S5: Structural characteristics of the *Lpa1-1* gene in maize genotypes and selected group of orthologue accessions; Table S6: *Zea mays* LPA1 protein (Protein ID: A7KVC2) domains and features.

**Author Contributions:** Conceptualization, V.M., F.H., K.K.P. and A.K.S.; Methodology, R.C., A.K. and V.R.; Software, A.K., S.R. and G.R.S.; Formal analysis, V.B.; Investigation, V.B.; Resources, F.H., V.M. and A.K.S.; Data curation, R.U.Z. and G.C.; Writing—original draft preparation, V.B. and V.M.; Writing—review and editing, F.H. and A.K.S.; Supervision, V.M., F.H. and K.K.P.; Project administration, F.H., V.M. and A.K.S.; Funding acquisition, F.H., V.M. and A.K.S. All authors have read and agreed to the published version of the manuscript.

**Funding:** Financial support from Indian Council of Agricultural Research (ICAR) sponsored Consortia Research Platform on "Molecular breeding for improvement of tolerance to biotic and abiotic stresses, yield and quality traits in crops-Maize component" (IARI Project Code No.: 12-143C), and ICAR-IARI, New Delhi is thankfully acknowledged.

**Institutional Review Board Statement:** Not applicable.

**Data Availability Statement:** The original contributions presented in the study are included in the article/Supplementary Materials, further inquiries can be directed to the corresponding author.

**Conflicts of Interest:** The authors declare no conflict of interest.

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
