# Peer review of "Molecular Characterization and Haplotype Analysis of Low Phytic Acid-1 (lpa1) Gene Governing Accumulation of Kernel Phytic Acid in Subtropically-Adapted Maize"

_agriculture, doi:10.3390/agriculture13071286_

Round 1

Reviewer 1 Report

This MS characterized the full-length Lpa1 gene sequence among three mutants and seven wild-type maize inbreeds and then conducted genetics, phylogenetic, physicochemical and structural analysis of the related sequences. Overall, this paper is generally well-written and provides fundamental data for further studies. I have no significant comments, only a few suggestions about L206-208. That is, the related information should be moved to Materials and methods section. Besides, why did you use Tajima-Nei method, by using jModelTest or other software?

Author Response

Response to reviewers’ comments

Reviewer 1:

This MS characterized the full-length Lpa1 gene sequence among three mutants and seven wild-type maize inbreeds and then conducted genetics, phylogenetic, physicochemical and structural analysis of the related sequences. 

Comment 1: Overall, this paper is generally well-written and provides fundamental data for further studies. I have no significant comments, only a few suggestions about L206-208. That is, the related information should be moved to Materials and methods section. 

Response 1: Thank you so much for your comments on the manuscript. As per the suggestion, we have shifted the related information of L-206-208 to the materials and methods section.

Comment 2: Besides, why did you use Tajima-Nei method, by using jModelTest or other software?

Response 2: Tajima-Nei distance gives a better estimate of the number of nucleotide substitutions than other softwares. Also, this assumes an equality of substitution rates among the sites and between transitional and transversional substitutions. Hence, the Tajima nei method is widely and most popularly used by several researches (Kubicek et al., 2019; Chhabra et al., 2021; Katral et al., 2022).

Reviewer 2 Report

Please find comments in attached find

Author Response

Response to reviewers’ comments

Reviewer 2:

Thank you very much for the critical suggestions and appreciations for improvement of the manuscript.

Comment 1: The conclusions are addressing the main question posed that of molecular characterization and haplotype analysis of low phytic acid gene that governs accumulation of kernel phytic acid. However, I feel the authors need to revisit the discussion as it appears there was a tendency of re-narrating the results instead of discussing the results vis a finding from other authors.

Response 1: Thank you very much for the constructive suggestion. The discussion part has been revised and focused to discuss on the results by citing most appropriate literature.

Reviewer 3 Report

In this study, the authors conducted a phylogenetic analysis to understand the development of several orthologues from a common ancestor. The study identified allelic variation in the Lpa1-1 gene in maize, with 42 haplotypes among 48 inbred varieties. The authors developed InDel markers that can characterize unknown genotypes for the lpa1 gene. Additionally, the study provided information on the LPA protein in maize and its orthologues. The authors observed variation in exon and intron lengths in the Lpa1 gene, which indicated evolutionary relationships among orthologues and maize. The findings of this study will be valuable for assessing genetic variation and advancing lpa1-based breeding programs in maize. This study is the first comprehensive investigation of the maize Lpa1 gene, its protein, and related orthologues.

Introduction: The introduction needs to be improved. The first paragraph of the introduction needs to be shortened. The focus of the introduction should be on Maize, phytic acid, and its importance, motivation, and impact of the current study, hypothesis, and objectives of the study. So, I suggest rewriting the introduction.

Line 218-219: Did you find the allele which is under selective pressure? Do you have any other evidence to support the statement that a particular allele is under selective pressure?

Section 3.8.: What is the source of the physicochemical properties?

Need to check for some minor grammatical and lexical errors.

Author Response

Response to reviewers’ comments

Reviewer 3:

In this study, the authors conducted a phylogenetic analysis to understand the development of several orthologues from a common ancestor. The study identified allelic variation in the Lpa1-1 gene in maize, with 42 haplotypes among 48 inbred varieties. The authors developed InDel markers that can characterize unknown genotypes for the lpa1 gene. Additionally, the study provided information on the LPA protein in maize and its orthologues. The authors observed variation in exon and intron lengths in the Lpa1 gene, which indicated evolutionary relationships among orthologues and maize. The findings of this study will be valuable for assessing genetic variation and advancing lpa1-based breeding programs in maize. This study is the first comprehensive investigation of the maize Lpa1 gene, its protein, and related orthologues.

Comment 1: Introduction: The introduction needs to be improved. The first paragraph of the introduction needs to be shortened. The focus of the introduction should be on Maize, phytic acid, and its importance, motivation, and impact of the current study, hypothesis, and objectives of the study. So, I suggest rewriting the introduction.

Response 1: Thanks for your valuable suggestion. We have revised the introduction and now it is more focussed on the points as per the suggestion.

Comment 2: Line 218-219: Did you find the allele which is under selective pressure? Do you have any other evidence to support the statement that a particular allele is under selective pressure?

Response 2: The value of Ka/Ks ratio was highest observed in that inbred.  Yes, we found the allele under selective pressure on the basis of significant diversity. The similar phenomenon was also observed in earlier reports in maize (Chhabra et al., 2021).

Comment 3: Section 3.8.: What is the source of the physicochemical properties?

Response 3:  The source of the physicochemical properties was ProtParam online tool (at ExPASy) in the section 3.8.

Comment 4: Need to check for some minor grammatical and lexical errors.

Response 4:  The manuscript has been thoroughly checked for grammatical and other lexical errors, as per the suggestion.